# DO LANGUAGE MODELS HAVE COMMON SENSE?

## ABSTRACT

It has been argued that current machine learning models do not have common sense, and therefore must be hard-coded with prior knowledge (Marcus, 2018). Here we show surprising evidence that language models can already learn to capture certain common sense knowledge. Our key observation is that a language model can compute the probability of any statement, and this probability can be used to evaluate the truthfulness of that statement. On the Winograd Schema Challenge (Levesque et al., 2011), language models are 11% higher in accuracy than previous state-of-the-art supervised methods. Language models can also be fine-tuned for the task of Mining Commonsense Knowledge on ConceptNet to achieve an F1 score of 0.912 and 0.824, outperforming previous best results (Jastrzebski et al., 2018). Further analysis demonstrates that language models can discover unique features of Winograd Schema contexts that decide the correct answers without explicit supervision.

## 1   INTRODUCTION

It has been argued that current machine learning models do not have common sense (Davis & Marcus, 2015; Marcus, 2018). For example, even best machine learning models perform poorly on commonsense reasoning tasks such as Winograd Schema Challenge (Levesque et al., 2011; Liu et al., 2016). This argument is often combined with another important criticism of supervised learning that it only works well on problems that have a lot of labeled data. The Winograd Schema Challenge (WSC) is an opposite of such problems because its labeled set size is only on the order of a few hundreds examples, with no official training data. Based on this argument, it is suggested that machine learning models must be integrated with prior knowledge (Marcus, 2018; Lenat, 1995).

As an example, consider the following question from the WSC dataset:

*"The trophy doesn't fit in the suitcase because **it** is too big."*
What is **"it"**? **Answer 0**: *the trophy*. **Answer 1**: *the suitcase*.

The main point of this dataset is that no machine learning model today can do a good job at answering this type of questions.

In this paper, we present surprising evidence that language models do capture certain common sense knowledge and this knowledge can be easily extracted. Key to our method is the use of language models (LMs), trained on a large amount of unlabeled data, to score multiple choice questions posed by the challenge and similar datasets. In the above example, we will first substitute the pronoun (*"it"*) with the candidates (*"the trophy"* and *"the suitcase"*), and then use an LM to compute the probability of the two resulting sentences (*"The trophy doesn't fit in the suitcase because **the trophy** is too big."* and *"The trophy doesn't fit in the suitcase because **the suitcase** is too big."*). The substitution that results in a more probable sentence will be the chosen answer. Using this simple method, we are able to achieve 63.7% accuracy, 11% above that of the previous state-of-the-art result[1].

To demonstrate a practical impact of this work, we show that the trained LMs can be used to enrich human-annotated knowledge bases, which are known to be low in coverage and expensive to expand. For example, *"Suitcase is a type of container"*, a relevant knowledge to the above Winograd Schema example, does not present in the ConceptNet knowledge base (Liu & Singh, 2004). The goal of this

---

[1]We open-sourced all language models used in this work. Links are excluded for anynomity.

task is to add such new facts to the knowledge base at a cheaper cost than human annotation, in our case using LM scoring. We followed the Commonsense Knowledge Mining task formulation from (Angeli & Manning, 2014; Li et al., 2016; Jastrzebski et al., 2018), which posed the task as a classification problem of unseen facts and non-facts. Without an additional classification layer, LMs are fine-tuned to give different scores to facts and non-facts tuples from ConceptNet. Results obtained by this method outperform all previous results, despite the small training data size (100K instances). On the full test set, LMs can identify commonsense facts with 0.912 F1 score, which is 0.02 better than supervised trained networks (Jastrzebski et al., 2018).

## 2 RELATED WORK

Previous attempts at solving Winograd Schema Challenge usually involve heavy utilization of annotated knowledge bases, rule-based reasoning, or hand-crafted features (Peng et al., 2015; Bailey et al., 2015; Schüller, 2014). Sharma et al. (2015) rely on a semantic parser to understand the question, query Google Search, and perform rule-based reasoning. Schüller (2014) formalizes the knowledge-graph data structure and a reasoning process based on cognitive linguistics theories. Bailey et al. (2015) introduce a mathematical reasoning framework with knowledge bases as axioms.

Rahman & Ng (2012) is an early empirical work towards WSC making use of learning. Their SVM, however, utilizes nearly 70K hand-crafted features and additional supervised training data, while being tested on a less restricted version of WSC. Concurrent work from Radford et al. (2018) attempts WSC by fine-tuning pretrained Transformer LMs on supervised training data, but did not produce better results than previous methods. In contrast, we make use of LSTMs, which are shown to be qualitatively different (Tang et al., 2018) and obtain significant improvements without fine-tuning.

The previous best method on WSC makes use of the skip-gram model to learn word representations (Liu et al., 2016). Their model, however, also includes supervised neural networks and three knowledge bases. Our work uses the same intuition that unsupervised learning from texts such as a skip-gram model can capture some aspect of commonsense. For example, Mikolov et al. (2013a;b) show that by learning to predict adjacent words in a sentence, word vectors can be used to answer analogy questions such as *Man:King::Woman:?*. The difference is that WSC requires more contextual information, and hence we use LMs instead of just word vectors. By training LMs on very large text corpora, we obtain good results without any supervised learning nor the aid of knowledge bases.

Closely related to our substitution method on Winograd Schema Challenge are Cloze type reading comprehension tasks such as LAMBADA (Paperno et al., 2016) or Store Cloze Test (Mostafazadeh et al., 2016), where LM scoring also reported great successes (Chu et al., 2016; Schwartz et al., 2017). On a broader impact, neural LMs have been applied to improve downstream applications (Dai & Le, 2015; Ramachandran et al., 2017; Peters et al., 2018; Howard & Ruder, 2018; Radford et al., 2018) by providing better sentence or paragraph vector representations.

Knowledge bases constructed by human are high in precision, but low in coverage. Since increasing the coverage by more human annotation is expensive, automated methods have been proposed. Previous attempts using deep neural networks are known to produce limited success on the ConceptNet knowledge base, where training data is limited. Li et al. (2016) shows that a supervised LSTM is outperformed by a simpler model in scoring unseen facts from ConceptNet. Furthermore, Jastrzebski et al. (2018) find Deep Neural Network's performance degrades significantly on a selected subset of most novel test instances in comparison to training data. In Section 5.2, we demonstrate that our trained LMs do not suffer from this phenomenon and outperform all previous methods on both test criteria.

## 3 METHODS

### 3.1 SUBSTITUTION FOR LM SCORING

In this section, we introduce a simple and straightforward application of pretrained language models on Winograd Schema Challenge. Our method is based on the observation that a language model can compute the probability of any given statement. We use this probability to judge the truthfulness of the statement.

We first substitute the pronoun in the original sentence with each of the candidate choices. The problem of coreference resolution then reduces to identifying which substitution results in a more probable sentence. Language modeling subsequently becomes a natural solution by its definition. Namely, language models are trained on text corpora, which encodes human knowledge in the form of natural language. During inference, LMs are able to assign probability to any given text based on what they have learned from training data. An overview of our method is shown in Figure 1.

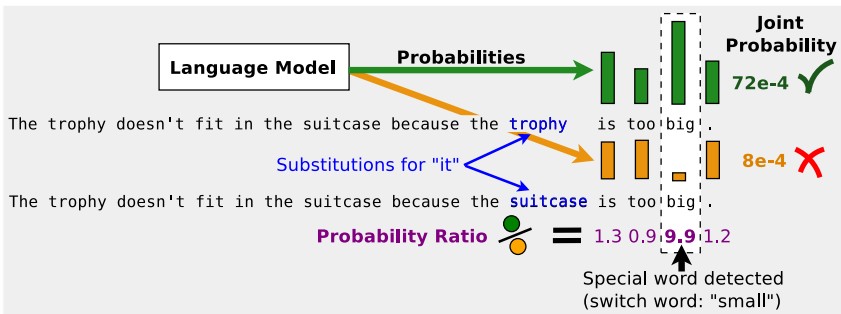

Figure 1: Overview of our method and analysis. We consider the test *"The trophy doesn't fit in the suitcase because it is too big."* Our method first substitutes two candidate references *trophy* and *suitcase* into the pronoun position. We then use an LM to score the resulting two substitutions. By looking at the probability ratio at every word position, we are able to detect *"big"* as the main contributor to *trophy* being the chosen answer. When *"big"* is switched to *"small"*, the answer changes to *suitcase*. This switching behaviour is an important feature characterizing the Winograd Schema Challenge.

Consider a sentence $S$ consisting of $n$ consecutive words has its pronoun to be resolved specified at the $k^{th}$ position:[2] $S = \{w_1, .., w_{k-1}, w_k \equiv p, w_{k+1}, .., w_n\}$. We make use of a trained language model $P_\theta(w_t | w_1, w_2, .., w_{t-1})$, which defines the probability of word $w_t$ conditioned on the previous words $w_1, ..., w_{t-1}$. The substitution of a candidate reference $c$ in to the pronoun position $k$ results in a new sentence $S_{w_k \leftarrow c}$ (we use notation $w_k \leftarrow c$ to mean that word $w_k$ is substituted by candidate $c$). We consider two different ways of scoring the substitution:

- $Score_{full}(w_k \leftarrow c) = P_\theta(w_1, w_2, ..., w_{k-1}, c, w_{k+1}, ..., w_n)$

which scores how probable the resulting full sentence is, and

- $Score_{partial}(w_k \leftarrow c) = P_\theta(w_{k+1}, ..., w_n | w_1, ..., w_{k-1}, c)$

which scores how probable the part of the resulting sentence following $c$ is, given its antecedent. In other words, it only scores a part of $S_{w_k \leftarrow c}$ conditioned on the rest of the substituted sentence. An example of these two scores is shown in Table 1. In our experiments, we find that the *partial* scoring strategy is generally better than the naive *full* scoring strategy. More comparison and analysis on scoring type is done in Section 6.3.

Table 1: Example of *full* and *partial* scoring for the test *"The trophy doesn't fit in the suitcase because it is too big."* with two reference choices *"the suitcase"* and *"the trophy"*.

| $c$ = the suitcase | $Score_{full}(w_k \leftarrow "the\ suitcase") = P(\text{The trophy doesn't fit in the suitcase because \textbf{the suitcase} is too big})$ |
|---|---|
| | $Score_{partial}(w_k \leftarrow "the\ suitcase") = P(\text{is too big} | \text{The trophy doesn't fit in the suitcase because \textbf{the suitcase}})$ |
| $c$ = the trophy | $Score_{full}(w_k \leftarrow "the\ trophy") = P(\text{The trophy doesn't fit in the suitcase because \textbf{the trophy} is too big})$ |
| | $Score_{partial}(w_k \leftarrow "the\ trophy") = P(\text{is too big} | \text{The trophy doesn't fit in suitcase because \textbf{the trophy}})$ |

## 3.2 RECURRENT LANGUAGE MODEL

We consider two types of Recurrent language models, one processes word inputs and the other processes character inputs. All output layers are constructed to only produce word outputs, allowing

---

[2]In Winograd Schema Challenge, $k$ is provided to avoid posing the question *"Who is him/her?"* or *"What is it?"*, exposing certain aspects of the correct answer.

both types of input processing to join in ensembles where the conditional probability at each word position is averaged over all ensemble members.

To handle human names in Winograd Schema Challenge, we simply make use of a very large vocabulary (approximately 800K tokens). We follow architectural design and training scheme in Józefowicz et al. (2016), with additional modifications to create more LM variants. More details about our LMs can be found in Appendix A.

## 4 EXPERIMENTAL SETTINGS

In this section we describe training text corpora used in our experiments. We also detail tests for commonsense reasoning and commonsense knowledge mining.

**Training text corpora.** We perform experiments on several different text corpora to examine the effect of training data type on test accuracy. Namely, we consider LM-1-Billion, CommonCrawl,[3] SQuAD and Gutenberg Books. For SQuAD, we collect context passages from the Stanford Question-Answering Dataset (Rajpurkar et al., 2016) to form its training and validation set accordingly.

**Commonsense Reasoning Tests.** We consider two tests: Pronoun Disambiguation Problems and Winograd Schema Challenge. The first consists of 60 pronoun disambiguation questions (PDP-60).[4] The latter consists of 273 questions and is designed to work against techniques such as traditional linguistic restrictions, common heuristics or simple statistical tests (Levesque et al., 2011).[5]

Rahman & Ng (2012) also built a Winograd Schema-like dataset but relaxed some criteria, allowing the context wording to reveal information about the correct answer.[6] We also found instances of incorrect annotation and ambiguous tests in their training and test sets (see Appendix C). In this work, therefore, we focus on the official Winograd Schema Challenge test set.

**Commonsense Knowledge Mining test.** Following (Angeli & Manning, 2014; Li et al., 2016), we use the same data split on the ConceptNet knowledge base, which results in training, validation and test sets having sizes of 100K, 1200, and 2400 respectively. With one half of the validation and test sets being non-facts, the commonsense knowledge mining task is posed as performing classification between facts and non-facts on these sets. Another test set in included which consists of 800 instances with highest novelty measurement computed against the training set (Jastrzebski et al., 2018).

## 5 MAIN RESULTS

We first train our LMs on all text corpora and test them on the two Commonsense Reasoning tests. The LMs are then finetuned for mining novel commonsense knowledge on ConceptNet.

### 5.1 COMMONSENSE REASONING TESTS

We first examine PDP-60 with unsupervised single-model resolvers by training one word-level LM on the Gutenberg corpus. In Table 2, this resolver outperforms the previous best result by more than 11% in accuracy. Next, we compare against systems that make use of both supervised and unsupervised training data. As can be seen in Table 2, the single-model LM can still produce better results when its competing system includes either supervised deep neural network or knowledge bases. By training more LMs for ensembling, we are able to reach 70% accuracy, outperforming the previous state-of-the-art result of 66.7%. For this task, we found *full* scoring gives better results than *partial* scoring. In Section 6.3, we provide more comparison between these two types of scoring.

---

[3]We evaluate all models trained on CommonCrawl after approximately 10 billion words are consumed.

[4]https://cs.nyu.edu/faculty/davise/papers/WinogradSchemas/PDPChallenge2016.xml

[5]https://cs.nyu.edu/faculty/davise/papers/WinogradSchemas/WS.html

[6]For example, in the schema *"The birds ate the seeds because they were hungry."*, *"hungry"* is only applicable to the correct answer *"the birds"* regardless of the context. In our WSC example, *"big"* is not particularly linked to either *suitcase* or *trophy*, requiring the system to make use of the context.

Table 2: Accuracy on PDP-60

| Method | Accuracy |
|---|---|
| Unsupervised Semantic Similarity Method (USSM) | 48.3 % |
| **Single-model LM-*full* (ours)** | **60.0 %** |
| USSM + Cause-Effect + WordNet (Miller, 1995) + ConceptNet (Liu & Singh, 2004) | 56.7 % |
| USSM + Supervised Deepnet | 53.3 % |
| USSM + Supervised Deepnet + 3 Knowledge Bases | 66.7 % |
| **Ensemble of 5 Unsupervised LMs-*full* (ours)** | **70.0 %** |

On the harder task WSC-273 where questions are designed to exclude relevant knowledge in their wording, incorporating supervised learning and knowledge base to USSM (Liu et al., 2016) provides insignificant gain this time (+3%), compared to the large gain on PDP-60 (+19%). On the other hand, our single-model resolver can still outperform the other methods by a large margin as shown in Table 3. By ensembling predictions from multiple LMs, we obtain nearly 10% of absolute accuracy improvement compared to the previous state-of-the-art. We note that Sharma et al. (2015) also attempted WSC but their approach is only applicable to 53 out of 273 test cases, therefore not comparable to our results.

Table 3: Accuracy on Winograd Schema Challenge

| Method | Accuracy |
|---|---|
| USSM + Knowledge Base | 52.0 % |
| USSM + Supervised DeepNet + Knowledge Base | 52.8 % |
| Single-model LM-*partial* | 56.4% |
| **Ensemble of 10 Unsupervised LMs-*partial*** | **61.5 %** |

**Customizing training data for Winograd Schema Challenge** As previous systems collect relevant data from knowledge bases after observing questions during evaluation (Rahman & Ng, 2012; Sharma et al., 2015), we also explored using this option. Namely, we build a customized text corpus based on questions in commonsense reasoning tasks. It is important to note that this does not include the answers and therefore does not provide supervision to our resolvers. In particular, we aggregate documents from the CommonCrawl dataset that have the most overlapping n-grams with the questions. The score for each document is a weighted sum of $F_1(n)$ scores when counting overlapping n-grams:

$$\text{Similarity\_Score}_{document} = \sum_{n=1}^{4} nF_1(n)$$

The top 0.1% highest ranked documents are chosen as our new training corpus. This procedure resulted in nearly 1,000,000 documents, with the highest ranking document having a score of $8 \times 10^{-2}$, still relatively small compared to a perfect score of $1.0$. We name this dataset *Stories* since most of the constituent documents take the form of a story with long chain of coherent events. More statistics on Stories can be found in Appendix B.

We train four different LMs on Stories and add them to the previous ensemble of 10 LMs, resulting in an accuracy of **63.7%** in the final system. Remarkably, single models trained on this corpus are already extremely strong, with one word-level LM achieving 62.6% accuracy.

## 5.2 MINING COMMONSENSE KNOWLEDGE WITH LM SCORING

In the previous sections, we show that unsupervised LMs can outperform other methods equipped with additional knowledge bases on two Commonsense Reasoning tests. In this section, we demonstrate how these trained LMs can help expand the coverage of these human-annotated knowledge bases.

To make LM scoring applicable, knowledge tuples of the form *Relation(head, tail)* from ConceptNet are first converted to a form that resembles natural language sentences. For example, *UsedFor(post*

*office, mail letter)* is converted to *"Post office is used for mail letter."* by simply concatenating its head, relation, and tail phrases in order. Although this simple procedure results in ungrammatical sentences,[7] we find our LMs can still adapt to this new data distribution and generalize extremely well to test instances.

For fine tuning, each commonsense fact in the training set is accompanied by a negative example, generated by replacing its tail phrase by another random phrase (for example *"Post office is used for dance."*). Instead of adding a classification layer, we add to the original LM objective a term that encourages perplexity discrepancy between the pair of positive and negative examples.

$$\text{Loss}_{new} = \text{Loss}_{LM} + \max(0, log(\text{Perp}_{positive}) - log(\text{Perp}_{negative}) + \alpha),$$

where $\alpha$ indicates how much of a discrepancy is needed beyond which no loss is added. $Perp$ is the perplexity evaluated on the tail phrase, given the corresponding head and relation phrases. During evaluation, a threshold is used to classify low-perplexity and high-perlexity instances as fact and non-fact. We found a word-level LM with $\alpha = 0.5$ perform best on the validation set.

Table 4: F1 scores on full test set proposed by Li et al. (2016) and novelty-based test set proposed by Jastrzebski et al. (2018).

| Method | Full test set | Novelty-based test set |
|---|---|---|
| DNN (Li et al., 2016) | 0.892 | 0.720 |
| Factorized (Jastrzebski et al., 2018) | 0.890 | 0.821 |
| Prototypical (Jastrzebski et al., 2018) | 0.794 | 0.574 |
| Single LM (ours) | **0.912** | **0.824** |

As shown in Table 4, our fine-tuned LM outperforms other methods on both tests. Unlike DNN (Li et al., 2016), LM ranking is robust to the novelty-based test instances, while supervised DNN performance degrades significantly on this test despite good performance on the full test. We suggest that this happened because supervised trained DNNs tend to overfit easily when training data is limited. On the other hand, by leveraging a massive amount of unsupervised training data, LM does not overfit to the limited training data for this task (100K instances) despite its large size of approximately 2 billion parameters.

# 6 ANALYSIS

In this section, we perform analysis on both correct and incorrect predictions made by LM scoring on the Winograd Schema, and the influence of training data types on test performance.

## 6.1 DISCOVERY OF WSC SPECIAL WORDS IN CORRECT PREDICTIONS

We introduce a method to detect keywords from the context at which our proposed resolvers make the decision between the two candidates $c_{correct}$ and $c_{incorrect}$. We then demonstrate that these detected keywords surprisingly match the annotated features in each Winograd Schema question that play the role of identifying the correct answer. Namely, we look at the following ratio:

$$q_t = \frac{P_\theta(w_t|w_1, w_2, ..., w_{t-1}; w_k \leftarrow c_{correct})}{P_\theta(w_t|w_1, w_2, ..., w_{t-1}; w_k \leftarrow c_{incorrect})},$$

where $1 \leq t \leq n$ for *full* scoring, and $k + 1 \leq t \leq n$ for *partial* scoring. It follows that the choice between $c_{correct}$ or $c_{incorrect}$ is made by whether the value of $Q = \prod_t q_t$ is bigger than 1.0 or not. By looking at the value of each individual $q_t$, it is possible to retrieve words with the largest values of $q_t$ and hence most responsible for the final value of $Q$.

We visualize the probability ratios $q_t$ to have more insights into the decisions of our resolvers. Figure 2 displays a sample of incorrect decisions made by *full* scoring which are corrected by *partial* scoring.

---

[7]A grammatical conversion should be *"The post office is used for mailing letters."*

Figure 2: A sample of questions from WSC-273 predicted incorrectly by *full* scoring, but corrected by *partial* scoring. Here we mark the correct prediction by an asterisk and display the normalized probability ratio $\hat{q}_t$ by coloring its corresponding word. It can be seen that the wrong predictions are made mainly due to $q_t$ at the pronoun position, where the LM has not observed the full sentence. *Partial* scoring shifts the attention to later words and places highest $q$ values on the special keywords, marked by a squared bracket. These keywords characterize the Winograd Schema Challenge, as they uniquely decide the correct answer. In the last question, since the special keyword appear before the pronoun, our resolver instead chose *"upset"*, as a reasonable switch word could be *"annoying"*.

Interestingly, we found $q_t$ with large values coincides with the special keyword of each Winograd Schema in several cases. Intuitively, this means our LMs assigned very low probability for the keyword after observing the wrong substitution. It follows that we can predict the keyword in each Winograd Schema question by selecting the word positions with the highest value of $q_t$.

Table 5: Accuracy of keyword detection from forward and backward scoring by retrieving top-2 tokens with the highest value of $q_t$

|  | Resolution accuracy | Special word retrieved |
|---|---|---|
| Forward scoring | 63.7% | 97 / 133 (70%) |
| Backward scoring | 58.2% | 18 / 45 (40%) |

For questions with keywords appearing before the reference, we detect them by backward-scoring models. Namely, we ensemble 6 LMs, each trained on one text corpus with word order reversed. Overall, we are able to discover a significant number of special keywords (115 out of 178 correctly answered questions) as shown in Table 5. This strongly indicates a correct understanding of the context and a good grasp of commonsense knowledge in the resolver's decision process.

## 6.2 WINOGRAD SCHEMA CONTEXT ABLATION

In the original proposal of the Winograd Schema Challenge, Levesque et al. (2011) argue that by careful wording of the context, no relevant knowledge is revealed about the correct answer. For example, *"big"* is not a property exclusive to either *"trophy"* or *"suitcase"*. This forces the system to resort to the context for correct answer, as opposed to exploiting simple statistical correlation in the training data to cheat the test.

We use the trained LMs to expose such correlation in the used training data by gradually ablating the context from a WSC question. For example, at 100% ablation, the scoring reduces to comparing only *"the trophy is too big"* versus *"the suitcase is too big"*. Figure 3-left shows that there is indeed such bias. For some LMs the bias made them perform worse than random at 100% ablation, while for others they perform better than random without any context. Note that this bias, however, does not necessarily affect the corresponding LM-scoring when context is included. As more context is included, all LMs improve and reach the best performance at 0% ablation, indicating the critical role of context in their scoring.

## 6.3 SCORING TYPE AND EFFECT OF TRAINING DATA.

We look at incorrect predictions from a word-level LM. With *full* scoring strategy, we observe that $q_t$ at the pronoun position is most responsible for a very large percentage of incorrect decisions as shown in Figure 2. For example, with the test *"The trophy cannot fit in the suitcase because **it** is*

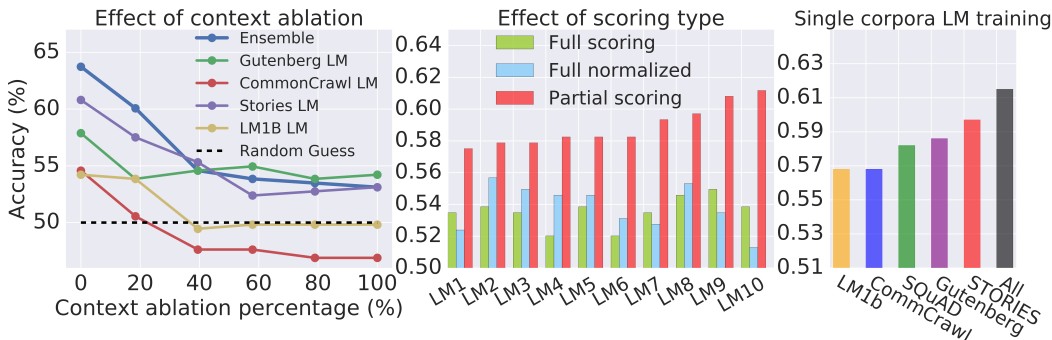

Figure 3: Analysis of different factors contributing to WSC test performance. **Left**: Context ablation with LMs trained on different text corpora. **Middle**: LM scoring type. **Right**: Training text corpus.

*too big.",* the system might return $c_{incorrect}$ =*"suitcase"* simply because $c_{correct}$ = *"trophy"* is a very rare word in its training corpus and therefore is assigned a very low probability, overpowering subsequent $q_t$ values.

To verify this observation, we apply a simple fix to *full* scoring by normalizing its score with the unigram count of $c$: $Score_{full\ normalized} = Score_{full}/Count(c)$. This normalization indeed fixes *full* scoring in 9 out of 10 tested LMs on PDP-60. On WSC-273, the observation is again confirmed as *partial* scoring, which ignores $c$ altogether, strongly outperforms the other two scorings in all cases as shown in Figure 3-middle. We therefore attribute the different behaviour observed on PDP-60 as an atypical case due to its very small size.

Next, we examine the effect of training data on commonsense reasoning test performance. An ensemble of 10 LMs is trained on each of the five corpora: LM-1-Billion, CommonCrawl, SQuAD, Gutenberg Books, and Stories. A held-out dataset from each text corpus is used for early stopping on the corresponding training data.[8] Figure 3-right shows that among single training text corpora, test performance improves as the training text contains longer documents (LM-1-Billion is a set of mostly independent sentences, while Gutenberg or Stories are full books or very-long documents). Finally, the ensemble trained on a mix of different datasets perform best, highlighting the important role of diversity in training data for commonsense reasoning accuracy of the final system.

## 7 CONCLUSION

We introduced a simple method to apply pretrained language models to tasks that require common-sense knowledge. Key to our method is the insight that large LMs trained on massive text corpora can capture certain aspect of human knowledge, and therefore can be used to score textual statements. On the Winograd Schema Challenge, LMs are able to achieve 11 points of accuracy above the best previously reported result. On mining novel commonsense facts from ConceptNet knowledge base, LM scoring also outperforms previous methods on two different test criteria. We analyse the trained language models and observe that key features of the context that identify the correct answer are discovered and used in their predictions.

Traditional approaches to capturing common sense usually involve expensive human annotation to build knowledge bases. This work demonstrates that commonsense knowledge can alternatively be learned and stored in the form of distributed representations. At the moment, we consider language modeling for learning from texts as this supplies virtually unlimited data. It remains an open question for unsupervised learning to capture commonsense from other modalities such as images or videos.

---

[8]To speed up training on these large corpora, we first train the models on the LM-1-Billion text corpus. Each trained model is then divided into three groups of parameters: Embedding, Recurrent Cell, and Softmax. Each of the three is optionally transferred to train the same architectures on CommonCrawl, SQuAD and Gutenberg Books. The best transferring combination is chosen on a held-out set.

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

## A    RECURRENT LANGUAGE MODELS

The base model consists of two layers of Long-Short Term Memory (LSTM) Hochreiter & Schmidhuber (1997) with 8192 hidden units. The output gate of each LSTM uses peepholes and a projection layer to reduce its output dimensionality to 1024. We perform drop-out on LSTM's outputs with probability 0.25.

Table 6: One-dimensional convolutional layers used to process character inputs

|                 | Conv 1 | Conv 2 | Conv 3 | Conv 4 | Conv 5 | Conv 6 | Conv 7 | Conv 8 |
|-----------------|--------|--------|--------|--------|--------|--------|--------|--------|
| Kernel size     | 1      | 2      | 3      | 4      | 5      | 6      | 7      | 7      |
| Output channels | 32     | 32     | 64     | 128    | 256    | 512    | 1024   | 2048   |

For word inputs, we use an embedding lookup of 800000 words, each with dimension 1024. For character inputs, we use an embedding lookup of 256 characters, each with dimension 16. We concatenate all characters in each word into a tensor of shape *(word length, 16)* and add to its two ends the *<begin of word>* and *<end of word>* tokens. The resulting concatenation is zero-padded to produce a fixed size tensor of shape *(50, 16)*. This tensor is then processed by eight different 1-D convolution (Conv) kernels of different sizes and number of output channels, listed in Table 6, each followed by a ReLU acitvation. The output of all CNNs are then concatenated and processed by two other fully-connected layers with highway connection that persist the input dimensionality. The resulting tensor is projected down to a 1024-feature vector. For both word input and character input, we perform dropout on the tensors that go into LSTM layers with probability 0.25.

Table 7: All variants of recurrent LMs used in our experiments.

| LM name    | Difference to base settings                                                       |
|------------|-----------------------------------------------------------------------------------|
| Word-LM 1  | Dropout rate 0.1                                                                   |
| Word-LM 2  | Learning rate 0.05                                                                 |
| Word-LM 3  | Residual connections around LSTM layers                                            |
| Word-LM 4  | Project dimension 2048, embedding dimension 2048, One layer of LSTM                |
| Char-LM 1  | Embedding dimension 4096, project dimension 2048                                   |
| Char-LM 2  | Embedding dimension 2048, project dimension 2048                                   |
| Char-LM 3  | Embedding dimension 1024, learning rate 0.1, Residual instead of Highway connection |
| Char-LM 4  | Learning rate 0.002, Embedding dimension 1024                                      |

We use a single fully-connected layer followed by a $Softmax$ operator to process the LSTM's output and produce a distribution over word vocabulary of size 800K. During training, LM loss is evaluated using importance sampling with negative sample size of 8192. This loss is minimized using the AdaGrad Duchi et al. (2011) algorithm with a learning rate of 0.2. All gradients on LSTM parameters and Character Embedding parameters are clipped by their global norm at 1.0. To avoid storing large matrices in memory, we shard them into 32 equal-sized smaller pieces. In our experiments, we used 8 different variants of this base model as listed in Table 7.

In Table 8, we listed all LMs and their training text corpora used in each of the experiments in Section 5.

## B    STORIES CORPUS SCORE RANKING

Figure 4 shows a histogram of similarity score introduced in Section 5.1. Inspecting an excerpt from the highest ranking document reveals many complex references from pronouns, within long chains of events. We hypothesize that this allows LM trained on this corpus to learn disambiguating pronouns to make correct predictions.

Table 8: Details of LMs and their training corpus reported in our experiments.

| Experiment | LM variant / training corpus |
|---|---|
| Single models on PDP-60 | Word-LM 1/Gutenberg and Char-LM 1/Gutenberg |
| Ensemble on PDP-60 | **Two single models on PDP-60 +** Word-LM 2/SQuAD + Char-LM 2/LM1B + Char-LM 3/CommonCrawl |
| Ensemble of 10 models on WSC-273 | **Ensemble on PDP-60 +** Word-LM 1/Gutenberg *(different random seed)* + Word-LM 1/LM1B + Char-LM 4/Gutenberg + Char-LM 4/SQuAD + Char-LM 4/CommonCrawl |
| Ensemble of 14 models on WSC-273 | **Ensemble of 10 models on WSC-273 +** Word-LM 1/Stories + Char-LM 2/Stories + Word-LM 3/Stories + Word-LM 4/Stories |
| Ensemble of 6 backward-scoring models on WSC-273 | Word-LM 1/Gutenberg + Word-LM 1/Stories + Char-LM 4/CommonCrawl + Char-LM 4/SQuAD + Word-LM 4/LM1B + Char-LM 2/Stories + |

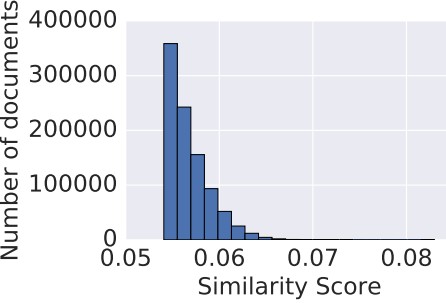

One day when John and I had been out on some business of our master 's , and were returning gently on a long , straight road , at some distance we saw a boy trying to leap a pony over a gate ; the pony would not take the leap , - and the boy cut him with the whip , but he only turned off on one side . He whipped him again , but the pony turned off on the other side . Then the boy got off and gave him a hard thrashing , and knocked him about the head ...

Figure 4: **Left**: Histogram of similarity scores from top 0.1% documents in CommonCrawl corpus, comparing to questions in Winograd Schema Challenge. **Right**: An excerpt from the document whose score is 0.083 (highest ranking). In comparison, a perfect score is of 1.0. Documents in this corpus contain long series of events with complex references from several pronouns.

## C    INCORRECT AND AMBIGUOUS ANNOTATIONS IN RELAXED WINOGRAD SCHEMA DATASET

On a non-exhaustive inspection of the dataset constructed by (Rahman & Ng, 2012), we found some instances of incorrect or ambiguous annotation.[9] Below we list two cases with our comment.

- A series of injections are used to battle a type of cancer in patients because **they** have a special type of drug which counteracts this sickness. **Label**: patients. **Comment**: Found in training set, incorrect label.

- John attacked Tim because **he** was a communist. **Label**: Tim. **Comment**: Found in test set, there is no clear answer to this question as communists can also attack their enemy.

## D    DATA CONTAMINATION IN COMMONCRAWL

Using the similarity scoring technique in section 5.1, we observe a large amount of low quality training text on the lower end of the ranking. Namely, these are documents whose content are mostly unintelligible or unrecognized by our vocabulary. Training LMs for commonsense reasoning tasks on full CommonCrawl, therefore, might not be ideal. On the other hand, we detected and removed a portion of PDP-122 questions presented as an extremely high ranked document.

---

[9]The released dataset can be found at `http://www.hlt.utdallas.edu/~vince/data/emnlp12/`, inspection done as of September 26th, 2018.

