# OpenReview forum: "Do Language Models Have Common Sense?"
_ICLR.cc/2019/Conference_

### Official Review · AnonReviewer3 · 2018-10-19
**Studying whether LM encode common-sense information. Novelty, clarity and methodology concerns**

**Rating:** 4
**Confidence:** 4

**Review:**

This paper experiments with pre-trained language models for common sense tasks such as Winograd Schema Challenge and ConceptNet KB completion. While the authors get high numbers on some of the tasks, the paper is not particularly novel, and suffers from methodology and clarity problems. These prevent me from recommending its acceptance.

This paper shows that pre-trained language models (LMs) can be used to get strong improvements on several datasets. While some of the results obtained by the authors are impressive, this result is not particularly surprising in 2018. In the last year or so, methods based on pre-trained LMs have been shown extremely useful for a very wide number of NLP tasks (e.g., Peters et al., 2018; Howard and Ruder, 2018; Radford et al., 2018). Moreover, as noticed to by the authors, Schwartz et al. (2017) demonstrated that LM perplexity can be useful for predicting common-sense information for the ROC story cloze task. As a result, the technical novelty in this paper is somewhat limited.

The paper also suffers from methodological problems:
-- The main results observed by the author, the large improvement on the (hard!) Winograd schema challenge, is questionable: The GLUE paper (Wang et al., 2018) reports that the majority baseline for this dataset is about 65%. It is unclear whether the authors here used the same version of the dataset (the link they put does not unambiguously decide one way or another). If so, then the best results published in the current paper is below the majority baseline, and thus uninteresting. If this is not the same dataset, the authors should report the majority baseline and preferably also run their model on the (hard) version used in GLUE.
-- The authors claim that their method on ConceptNet is unsupervised, yet they tune their LM on triplets from the training set, which makes it strongly rely on task supervision.

Finally, the paper suffers clarity issues.
-- Some sections are disorganized. For instance, the experimental setup mentions experiments that are introduced later (the ConceptNet experiments).
-- The authors mention two types of language models (word and character level), and also 4 text datasets to train the LMs on, but do not provide results for all combinations. In fact, it is unclear in table 2 what is the single model and what are the ensemble (ensemble of the same model trained on the same dataset with different seeds? or the same model with different datasets?).
-- The authors do not address hyper-parameter tuning.
-- What is the gold standard for the "special word retrieved" data? how is it computed?


Other comments:
-- Page 2: "In contrast, we make use of LSTMs, which are shown to be qualitatively different (Tang et al., 2018) and obtain significant improvements without fine-tuning.": 1. Tang et al. (2018) do not discuss fine-tuning. 2. Levy et al. (ACL 2018) actually show interesting connections between LSTMs and self-attention.
-- Schwartz et al. (2017) showed that when using a pre-trained LM, normalizing the conditional probability of p(ending | story) by p(ending) leads to much better results than  p(ending | story). The authors might also benefit from a similar normalization.
-- Page 5: how is F1 defined?

Minor comments:
-- Page 2: " ... despite the small training data size (100K instances).": 100K is typically not considered a small training set (for most tasks at least)
-- Page 5: "... most of the constituent documents ...": was this validated in any way? how?
-- The word "extremely" is used throughout the paper without justification in most cases.


Typos and such:
page 1: "... a relevant knowledge to the above Winograd Schema example, **does** not present ... ": should be "is"
page 5: "In the previous sections, we ***show*** ...": showed
page 7: "For example, with the ***test*** ...": "test instance"

---

### Official Review · AnonReviewer1 · 2018-11-02
**Two somewhat disconnected small contributions**

**Rating:** 4
**Confidence:** 4

**Review:**

This paper uses a language model for scoring of question answer candidates in the Winograd schema dataset, as well as introduces a heuristic for scoring common-sense knowledge triples.

Quality:
Pros: The paper shows improvements over previous papers for two tasks related to common-sense knowledge. They both mainly utilise simple language models, which is impressive. The second one uses an additional supervised collaborative filtering-style model. The authors further perform a detailed error analysis and ablation study.
Cons: The paper isn't very well-written. It contains quite a few spelling mistakes and is unclear in places. The Winograd Scheme Challenge isn't a very interesting dataset and isn't widely used. In fact, this is evidenced by the fact that most cited papers on that datasets are preprints and technical reports.

Clarity:
The paper is confusing in places. It should really be introduced in the abstract what is meant by "common sense". Details of the language model are missing. It is only clear towards the end of the introduction that the paper explores two loosely-related tasks using language models.

Originality:
Pros: The suggested model outperforms others on two datasets.
Cons: The suggested models are novel in themselves. As the authors also acknowledge, using language models for scoring candidates is a simple baseline in multiple-choice QA and merely hasn't been tested for the Winograd schema dataset.

Significance:
Other researchers within the common-sense reasoning community might cite this paper. The significance of this paper to a larger representation learning audience is rather small.

---

### Official Review · AnonReviewer2 · 2018-11-02
**some interesting results, but could use more rigor and empirical exploration**

**Rating:** 5
**Confidence:** 4

**Review:**

This paper evaluates language models for tasks that involve "commonsense knowledge" such as the Winograd Schema Challenge (WSC), Pronoun Disambiguation Problems (PDP), and commonsense knowledge base completion (KBC).

Pros:

The approach is relatively simple in that it boils down to just applying language models.

The results outperform prior work, in some cases by pretty large margins.

The language models are quite large and it appears that this is the first time that large-scale language models have been applied seriously to the Winograd Schema Challenge (rather than, say, to the NLI version of it in GLUE, to which it is hard to compare these results).

Some of the additional and ablation experiments are interesting.


Cons:

While this paper has some nice results, there are some aspects of it that concern me, specifically related to hyperparameter tuning and experimental rigor:

There are three methods given for using an LM to make a prediction: full, full-normalized, and partial. For PDP, full (or perhaps full-normalized?) works best, while for WSC, partial works best. The differences among methods, at least for WSC, are quite large: from 2% to 10% based on Figure 3. I don't see a numerical comparison for PDP, so I'm not sure how these methods compare on it. Since the datasets are so small, there is no train/dev/test split, so how were these decisions made? They seem to be oracle decisions. This is concerning to me, as there is not much explanation given for why one method is better than another method.

My guess is that the reason why partial works better than full for WSC is because the WSC sentences were constructed such that the words up to and including the ambiguous pronoun were written such that it would be difficult to identify the antecedent of the pronoun. The rest of the sentence would be needed to identify the antecedent. I'll assume for this discussion that the sentence can be divided into three parts x, y, and z, where x is the part before the pronoun, y is the phrase that replaces the pronoun, and z is the part after the pronoun. Then p(z|xy), which is partial scoring, corresponds to p(xyz)/p(xy), which can be viewed as "discounting" or "normalizing for" the probability of putting y in place of the pronoun given the context x. For WSC, I think one of the goals in writing the instances is to make the "true" p(xy) approximately equal for both values of y. The language model will not naturally have this be the case (i.e., that p(xy) is the same for both antecedents), so dividing by p(xy) causes the resulting partial score to account for the natural differences in p(xy) for different antecedents. This could be explored empirically. For example, the authors could compute p(xy) for both alternatives for all PDP and WSC instances and see if the difference (|p(xy_1) - p(xy_2)|, where y_1 and y_2 are the two alternatives) is systematically different between WSC and PDP. Or one could see if p(xy) is greater for the antecedent that is closer to the pronoun position or if it is triggered by some other effects. It could be the case that the PDP instances are not as carefully controlled as the WSC instances and therefore some of the PDP instances may exhibit the situation where the prediction can be made partially based on p(xy). The paper does not give an explanation for why full scoring works better for PDP and chalks it up to noise from the small size of PDP, but I wonder if there could be a good reason for the difference.

The results on KBC are positive, but not super convincing. The method involves fine-tuning pretrained LMs on the KBC training data, the same training data used by prior work. The new result is better than prior work (compared to the "Factorized", the finetuned LM is 2.1% better on the full test set, and 0.3% better on the novelty-based test set), but also uses a lot more unlabeled data than the prior work (if I understand the prior work correctly). It would be more impressive if the LM could use far fewer than the 100K examples for fine-tuning. Also, when discussing that task, the paper says: "During evaluation, a threshold is used to classify low-perplexity and high-perlexity instances as fact and non-fact." How was this threshold chosen?

I also have a concern about the framing of the overall significance of the results. While the results show roughly a 9% absolute improvement on WSC, the accuracies are still far from human performance on the WSC task. The accuracy for the best pretrained ensemble of LMs in this paper is 61.5%, and when training on WSC-oriented training data, it goes up to nearly 64%. But humans get at least 92% on this task. This doesn't mean that the results shouldn't be taken seriously, but it does suggest that we still have a long way to go and that language models may only be learning a fraction of what is needed to solve this task. This, along with my concerns about the experimental rigor expressed above, limits the potential impact of the paper.


Minor issues/questions:

In Sec. 3.1: Why refer to the full scoring strategy as "naive"? Is there some non-empirical reason to choose partial over full?

The use of SQuAD for language modeling data was surprising to me. Why SQuAD? It's only 536 articles from Wikipedia. Why not use all of Wikipedia? Or, if you're concerned about some of the overly-specific language in more domain-specific Wikipedia articles, then you could restrict the dataset to be the 100K most frequently-visited Wikipedia articles or something like that.

I think it would be helpful to give an example from PDP-60.

Sec. 5.1: How is F_1(n) defined?  I also don't see how a perfect score is 1.0, but maybe it's because I don't understand how F_1(n) is defined.

Sec. 6.1: Why would t range from 1 to n for full scoring? Positions before k are unchanged, right? So q_1 through q_{k-1} would be the same for both, right?

In the final example in Figure 2, I don't understand why "yelled at" is the keyword, rather than "upset". Who determined the special keywords?

I was confused about the keyword detection/retrieval evaluation. How are multi-word keywords handled, like the final example in Figure 2? The caption of Table 5 mentions "retrieving top-2 tokens". But after getting the top 2 tokens, how is the evaluation done?

Sec. 6.3 says: "This normalization indeed fixes full scoring in 9 out of 10 tested LMs on PDP-60." Are those results reported somewhere in the paper? Was that normalization used for the results in Table 2?

Sec. 6.3 says: "On WSC-273, the observation is again confirmed as partial scoring, which ignores c [the candidate] altogether, strongly outperforms the other two scorings in all cases" -- What is meant by "which ignores c altogether"?  c is still being conditioned on and it must not be ignored or else partial scoring would be meaningless (because c is the only part that differs between the two options).


Typos and minor issues:

Be consistent about "common sense" vs. "commonsense".

Be consistent about "Deepnet" vs. "DeepNet" (Tables 2-3).

Sec. 1:
"even best" --> "even the best"
"such as Winograd" --> "such as the Winograd"
"a few hundreds" --> "a few hundred"
"this type of questions" --> "this type of question"
"does not present" --> "is not present"
"non-facts tuples" --> "non-fact tuples"

Sec. 2:
"solving Winograd" --> "solving the Winograd"
"Store Cloze" --> "Story Cloze"
"constructed by human" --> "constructed by humans"

Sec. 4:
What is "LM-1-Billion"?
Why SQuAD?
"Another test set in included" --> "Another test set is included"

Sec. 5.2:
Check margin in loss_new

"high-perlexity" --> "high-perplexity"

Sec. 6:
Figure 2 caption: "keyword appear" --> "keyword appears"

Sec. 6.2:
"for correct answer" --> "for the correct answer"

Appendix A:
"acitvation" --> "activation"
Appendix B:
Figure 4 caption: "is of" --> "is"
The right part of Figure 4 has some odd spacing and hyphenation.

---

### Public Comment · (anonymous) · 2018-10-08
**Model Selection**

Thanks for the work.

Concerning the table 7 and table 8 in the appendix, it seems to me that you have 8 LM variations for each corpus which represents 40 possible single LM models. When you ensemble the choice is not only an arbitrary subset of 14 of them, but involves combinations of these LM variations that are not at all consistent (for example, why do you use LM-2 on SQUAD and not LM-1?). Did you use any auxiliary task to do the model selection? If yes, I think it should be added to the paper.

---

> ### Author Response · Authors · 2018-10-08
> **Models are chosen based on validation perplexity**
>
> Hi, thanks for the question. Below we include all details throughout our experiments to answer your question as well as any other potential inquiries about the training process.
>
> TLDR: First Heuristic =  training corpus diversity (see section 6.3 and Figure 3-right for relevant analysis), Secondary heuristic = validation perplexity on corresponding held-out data.
>
> Ensemble choice is made to first include as many corpora as possible (Section 6.3 and Figure 3-Right show relevant analysis):
>
> * For single models on PDP-60 we chose Gutenberg as this is also the training corpus used in the previous SOTA [1]. Single model Char-LM result is not included to avoid complicating the tables, but its performance is also better than USSM (53.5%).
> * For ensemble of 5, we simply add all 3 of the remaining datasets (hence 1 LM each).
> * For ensemble of 10, we repeat the previous corpora choice twice.
> * For ensemble of 14, we add 4 LMs from Stories.
>
> Once the training corpus profile is decided, we train and chose LMs based on perplexity on a held-out set. One such held-out set is constructed for each training data, as opposed to a single joint held-out set for all training corpora, since later on we want to demonstrate the effect of training corpus choice on commonsense reasoning test performance.
>
> Note that we did not construct and train Word-LM-4 and Char-LM-4 until Section 5.2 (evaluation on Winograd Schema Challenge). There is no particular reason besides we want to push ensemble performance for better results by adding more LMs (even though single models are already better than previous results, see Table 3).
>
> Not all of our LMs converged to a good perplexity (below 40 points) on the corresponding validation sets, some other LMs diverged (perplexity > 100), we discarded those models. We initially choose learning rate 0.2 following [2] and randomly try some other learning rates for wordLM2, charLM3 and charLM4 since some of them diverged on LM1B (see footnote 8). Those learning rate are finally fixed and used on all subsequent datasets (CommonCrawl, SQuAD, Gutenberg, and Stories), which is why not all LM-corpus pairs work out in the end.
>
> Other than trying out some learning rate values above, we did not perform any tuning since it takes time (on average, an LM took at least 1 million steps for its held-out perplexity to stop improving, which amounts to approximately 01 month of training on a Tesla P-100 GPU), and we already obtain good results.
>
> [1] Quan Liu, Hui Jiang, Zhen-Hua Ling, Xiaodan Zhu, Si Wei, and Yu Hu. Combing context and
> commonsense knowledge through neural networks for solving winograd schema problems. CoRR,
> abs/1611.04146, 2016.
>
> [2] Rafal Józefowicz, Oriol Vinyals, Mike Schuster, Noam Shazeer, and Yonghui Wu. Exploring the
> limits of language modeling. CoRR, abs/1602.02410, 2016.

---

> > ### Public Comment · (anonymous) · 2018-10-09
> > **Ensembling**
> >
> > Hi, Thanks for your answer. My question was not about the choice of parameters for the single LM, which we agree seems to outperform USSM and random baseline.
> >
> > Instead, my question was about model selection when you ensemble. Consider that I have 40 random classifiers for the WSC; if I choose 14 of them based on the accuracy on the the WSC (test set), it's really likely that I get good results.
> >
> > You have 5 corpora (LM1b, SQUAD, CommCrawl, Gutenberg and Stories) and 8 different LM settings (ranging between hyperparameters that differ from the base settings as well as choice of word-level vs character level). This amounts to 40 possible LMs, of which you choose 14. This number (14) is a hyperparameter in itself. How did you come up with it? In addition, when you do model selection to find an ensemble of 10 models (in the case of not using Stories) you use 4 different LMs trained on Gutenberg (2  Word-LM1’s with different random seeds as well as a one Char-LM4 and one Char-LM 1), but only two on CommonCrawl (Char-LM 4, Char-LM 3). Obviously, the choice seems inconsistent and it does not seem to be based on validation perplexity. Otherwise, why would you use the same model two times?
> >
> > Could you be a bit more clear on how you selected the models?

---

> > > ### Author Response · Authors · 2018-10-09
> > > **I see your point, validation perplexity is what we used.**
> > >
> > > > This amounts to 40 possible LMs, of which you choose 14. This number (14) is a hyperparameter in itself.
> > >
> > > I see what you mean, which we have already addressed in the previous answer. We do not train a very large number of LMs and then tune the ensemble size or selection. We gradually train and add new LMs to the ensemble up to 10 LMs originally (Section 5.1) and observe diminishing returns, so we push further using another direction (Section 5.2) of customizing data.
> > >
> > > Why stop at 14? We have very large ensembles that achieve only slightly better results (64.4\%), which is not meaningful as 64% accuracy is still very far from human level accuracy. Besides, single model on Stories has already achieved 62.6\% accuracy.
> > >
> > > The point of our paper is proving that LMs can perform better than previous methods, and we demonstrated two ways to improve upon single-LM (ensembling and customizing training data). If one tune the LMs more we believe 70\% is achievable, but 80\% or above will need something entirely different, but that is entirely speculative.
> > >
> > > > you use 4 different LMs trained on Gutenberg ... but only two on CommonCrawl.
> > >
> > > As noted above, we do not decide the total number of LMs before hand, but add new LMs as experiments go:
> > >
> > > The original 2 LMs on Gutenberg are Word-LM1 and Char-LM1. Gutenberg is used as it is used to trained USSM. With 66.7\% of USSM + knowledge bases + supervised deep net, single LM are now far behind (60\%) and we started to explore ensembling with other training corpora.
> > >
> > > The ensemble of 5 is just adding SQuAD, LM1T, LM1B. The ensemble of 10 is just doubling the choice from the previous ensemble of 5, leading to 4 on Gutenberg and 2 on all other datasets. Doubling from 5 to 10 is a simple and obvious choice to us, albeit somewhat arbitrary. A different choice could result in better or worse, but is likely to improve upon ensemble of 5, which will also support our method of ensembling.
> > >
> > > > Obviously, the choice seems inconsistent and it does not seem to be based on validation perplexity. Otherwise, why would you use the same model two times?
> > >
> > > It is clearly not the same model twice, since they started from different initialization. Our training of LMs is full of models that failed to converge, implementation debugging, transferring pretrained parameters (footnote 8), so it might not be as clean as one wish to see from the first glance. We tried our best to summarize necessary details in Appendix A and the above comment. Thanks for going through them in details.

---

> > > > ### Public Comment · (anonymous) · 2018-10-09
> > > > **Re-wording OP's question.**
> > > >
> > > > Hi there,
> > > >
> > > > I believe the original poster has raised an important question about your paper, and I agree that you are not directly answering his question. Repeating results like 64.4%, 62.6, potential for 70% has nothing at all to do with the important question of *model selection*. You say you "gradually train and add new LMs to the ensemble up to 10 LMs originally". However, the question is, what made you choose *certain* LM's over others at each step. For example, you would choose to add an LM-2 which would vary from the base setting in terms of some hyperparameter, and then suddenly jump (seemingly arbitrarily) to another LM choice, say, char LM-4. What drove these arbitrary choices? Was it a greedy process on the Winograd Schema Challenge accuracy? I understand that you say that *each* LM had a validation perplexity to it; is this what you used to choose certain LM's over others? And if that is the case, I'm surprised that none of these details were included in the paper (as well as even mentioning that the validation perplexity was used). Ultimately, when you consider that you "gradually" ensembled on WSC, which is a *test set*, and observed accuracies en route, this is precisely antithetical with the purpose of a test set.

---

### Public Comment · (anonymous) · 2018-10-16
**Reproducibility**

Hi,

Thanks for your work ! Using your code available on Github, I tried to reproduce the results on the Winograd Schema Challenge. Regarding the ensemble of 10LMs and the ensemble of 14LMs, I get a similar accuracy (61.5% and 63.7% accuracy). However, regarding the performance of the single LM, I don't get the same accuracy. I have the following results:

Model |  LM1  |  LM2  |  LM3  |  LM4  |  LM5  |  LM6  |  LM7  |  LM8  |  LM9  |  LM10  |  LM11  |  LM12  |  LM13  | LM14
-------------------------------------------------------------------------------------------------------------------------------------------------------------------
Acc.     |54.6% |50.2% |54.2%  |55.0% |54.2% |55.0% | 55.3% | 56.8%|57.9% | 57.5%   | 55.7%  | 58.2%   | 60.8% | 56.0%

The results clearly show that the performance of the single LM is not random and that they capture patterns that are useful for the task. However, I don't understand what is the accuracy reported in table 3, 56.4% for a single LM and the accuracy reported at the end of paragraph 5.1 'with one word level LM achiving 62.6% accuracy'. Could you comment on that ?

---

> ### Author Response · Authors · 2018-10-29
> **Re: Reproducibility**
>
> Hi! Thank you for using our code and report your results here. It seems some numbers from the table are different than what we had and the latest release of Tensorflow indeed produces those number. We are checking if there is a mismatch in terms of software or the language model version. Either way, we will make updates so reported results match with the open-source release.

---

### Meta-Review · Area_Chair1 · 2018-12-16
**Clarity and Evaluation Issues**

**Confidence:** 4
**Recommendation:** Reject

**Metareview:**

This paper adapts language models (LMs), recurrent models trained on large corpus to produce the next word in English, to two commonsense reasoning tasks: the Winograd schema challenge and commonsense knowledge extraction. For the former, the language model score itself is used to obtain substantial gains over existing approaches for this challenging task, while a slightly more involved training procedure adapts the LMs to commonsense extraction. The reviewers appreciated the simplicity of the changes to existing LMs and the impressive results (especially on the WSC).

The reviewers point out the following potential weaknesses: (1) clarity issues in the writing and the presentation, (2) a lack of novelty in the proposed approach, given a number of recent work has shown the ability of language models to perform commonsense reasoning, and (3) critical methodological issues in the evaluation that raise questions about the significance of the results. A lack of response from the authors meant that there was no further discussion needed, and the reviewers encourage the authors to take the feedback to improve further versions of the paper.